

# The impact of seismic noise produced by wind turbines on seismic borehole measurements

Fabian Limberger[1,2], Georg Rümpker[1,3], Michael Lindenfeld[1], and Hagen Deckert[2]

[1]Institute of Geosciences, Goethe-University Frankfurt, 60438 Frankfurt am Main, Germany
[2]Institute for Geothermal Resource Management (igem), 55411 Bingen, Germany
[3]Frankfurt Institute for Advanced Studies (FIAS), 60438 Frankfurt am Main, Germany

*Correspondence to:* Fabian Limberger (limberger@igem-energie.de)

**Abstract.** Seismic signals produced by wind turbines can have an adverse effect on seismological measurements up to distances of several kilometres. Based on numerical simulations of the emitted seismic wavefield, we study the effectivity of seismic borehole installations as a way to reduce the incoming noise. We analyse the signal amplitude as a function of sensor depth and investigate effects of seismic velocities, damping parameters and geological layerings in the subsurface. Our numerical approach is validated by real data from borehole installations affected by wind turbines. We demonstrate that a seismic borehole installation with an adequate depth can effectively reduce the impact of seismic noise from wind turbines in comparison to surface installations. Therefore, placing the seismometer at greater depth represents a potentially effective measure to improve or retain the quality of the recordings at a seismic station. However, the advantages of the borehole decrease significantly with increasing signal wavelength.

## 1 Introduction

Global warming, energy crises and hence the goal to reduce the dependency on fossil energies demonstrate the relevance of exploiting renewable energies, including wind power. Thus, the increase of the number of wind turbines (WTs) plays a key role in the coming years. WTs are preferably installed in remote areas with windy conditions to increase the power production and to minimize their impacts (e.g., shadowing, acoustic noise and infrasound) on the environment. Seismic stations are often located in areas with similar conditions due to the low seismic noise levels compared to urban areas with anthropogenic noise sources as traffic, industry and railways. Nevertheless, the vibration of WTs can have a significant impact on seismic stations and networks. However, the effects strongly depend on the distance between the seismometer and the WT.

Seismic signals of WTs are characterized by frequencies between 1 Hz and 10 Hz and have been described in detail in a number of studies (e.g., Saccorotti et al., 2011; Stammler and Ceranna, 2016; Zieger and Ritter, 2018). The systematic decays of the corresponding signal amplitudes with distance from the WT or wind farms (WFs) have been analysed at various WFs (Neuffer and Kremers, 2017; Limberger et al., 2021; Gassner et al. 2022). Analytical and numerical approaches to model the amplitudes have been developed in terms of considering single WTs (Gortsaset al., 2017; Lerbs et al., 2020; Abreu et al., 2022)



and complete WFs (Limberger et al., 2021, 2022) including wavefield interferences from multiple WTs. On this basis, methods for predicting and reducing seismic noise from WTs or other noise sources are developed taking into account, e.g., meta materials (Colombi et al., 2016; Abreu et al., 2022), interferences and topographic effects (Limberger et al., 2021, 2022) and denoising methods (e.g., Heuel and Friederich, 2022). However, effective and robust solutions to compensate the seismic noise without losing the quality of the natural seismological signals are missing. It is generally known that seismometers in boreholes have lower noise levels compared to stations at the surface (Withers et al., 1996; Boese et al., 2015) which can improve the detectability of seismic events even in urban areas (Malin et al. ,2018). Boese et al. (2015) reported a noise level reduction of up to 30 dB (average 10 dB) on a 383 m deep borehole sensor compared to a surface sensor for frequencies $\geq$ 1 Hz. Similar effects of borehole installations on signals from WTs are shown by Zieger and Ritter (2018). They compared signals measured in boreholes with surface data and showed a significant reduction of the surface wave amplitude induced by a nearby WF. Neuffer and Kremers (2017) analysed data from borehole stations as well, but did not systematically study the relation to surface data. Nevertheless, they estimated a noise reduction by an order of magnitude due to the borehole installation. Obviously, borehole installations can play a relevant role in reducing the noise of WTs at seismometers. However, their capabilities, limitations and the predictability of its effectivity has not been studied in detail.

Here, we investigate the effectivity of borehole installations using numerical simulations. We perform sensitivity studies in view of signal frequencies, seismic velocities, homogeneous and layered subsurface structures, attenuation and the distance between source and receivers on depth-dependent signal amplitudes. We compare our numerical results with data from borehole measurements reported by Zieger and Ritter (2018). Our results provide constraints on the distances between WT and seismic stations necessary to reduce the noise levels to a desired level.



## 2 Model setup and data processing

### 2.1 Description of the numerical model

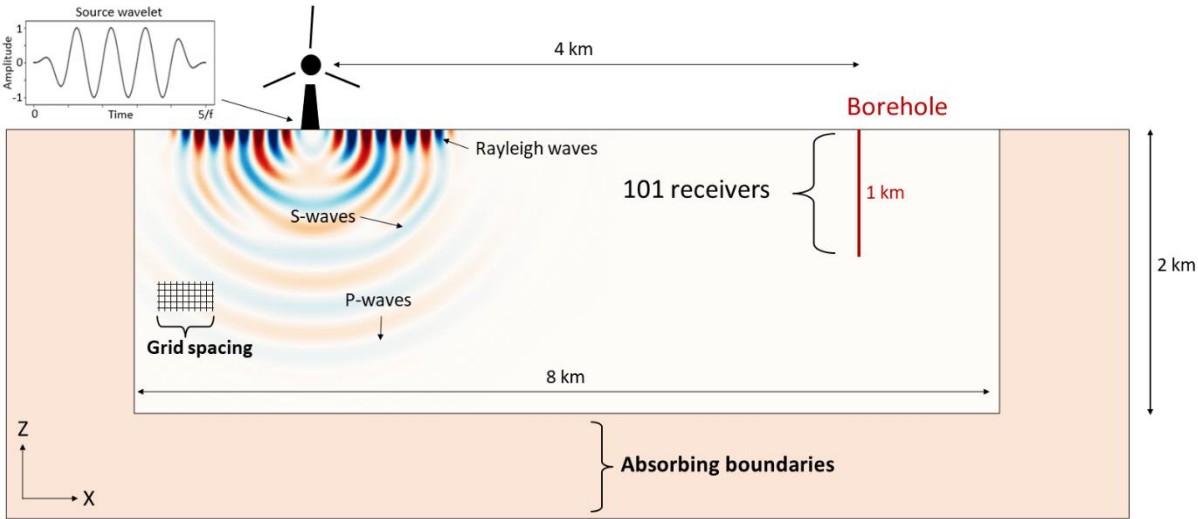

**Figure 1:** The numerical model includes a sinusoidal source wavelet, receivers located along a line from surface to a depth of 1 km and a sufficient grid spacing (three elements per minimum wavelength of the simulation) as well as absorbing boundaries (two times the maximum wavelength of the simulation). P-waves, S-Waves and surface waves are simulated during the forward modelling. Synthetic seismograms are extracted at positions indicated by the red line (borehole).

The forward modelling of the wave propagation is performed in two dimensions (x-z plane) using the software package Salvus (Afanasiev, et al., 2018), which enables the simulation the complete wavefield (P-wave, S-wave and surface waves, including conversion and scattering effects). A comparison of the results with a simulation in three dimensions shows that a two-dimensional approach seems sufficient for addressing the described problem (the corresponding data are in the supplements (Fig. S1)). The seismic source is located at the surface of the model domain (Fig. 1). The source wavelet is a tapered sinusoidal function with a length of five signal periods, which implies that the source duration increases for simulations with lower frequencies. The exciting force is assumed to be vertically oriented. The modelling domain has a length of 8 km (x-direction) and a depth of 2 km. Absorbing boundaries are added to all sides, except for the free surface on top of the model. The absorbing boundary has a minimum thickness of two times the maximum wavelength used during the simulation to sufficiently suppress reflections at the sides. A synthetic 1-km deep borehole is located in a distance of 4 km from the source. Receivers are located at intervals of 10 m along the borehole to extract the synthetic seismograms at 101 positions. In this work, we study the effects of both homogeneous and layered models including effects of varying seismic velocities. The velocity of the P-wave is calculated from $V_P=1.7 \times V_S$ and the density is 2600 kg m$^{-3}$ in every simulation. The source frequency is systematically increased from 0.2 Hz to 6 Hz (step size 0.2 Hz) to cover a wide range of typical signal frequencies observed for WTs (see





references). A separate simulation is performed for each frequency and model. The grid spacing is generated using three elements per minimum wavelength to avoid numerical artefacts. All studied models are listed in Table 1. Models 1-9 are used to study general effects of seismic velocities, geological layerings and attenuation. Model 10 is generated based on results from

70 the MAGS2 project (Spies et al., 2017), which provided detailed information on the seismic velocities in the region of Landau in Rhineland-Palatinate, Germany. We use this information about the local subsurface to establish a corresponding average velocity model (Fig. S2) and to perform the real data validation of our proposed solutions.

**Table 1:** List of models used in this study. Models 1-9 are generic to study effects of geophysical parameters and layers in the

75 subsurface. Model 10 is used for the validation with real data. The quality factor Q describes the loss of energy per seismic wave cycle due to anelastic processes or friction inside the rock during the wave propagation. The damping of the P-wave and S-wave is decreasing with increasing $Q_P$ and $Q_S$.

| ID | DESCRIPTION | $V_{S1}$ | $V_{S2}$ | $V_{S3}$ | $V_{S3}$ | $V_P$ |
|---|---|---|---|---|---|---|
| Model1 | Homogeneous half space | 500 | - | - | - | |
| Model2 | Homogeneous half space | 1000 | - | - | - | |
| Model3 | Homogeneous half space | 1500 | - | - | - | |
| Model4 | Two layers (z=-200m), low velocity | 500 | 1000 | - | - | 1.7 $V_S$ |
| Model5 | Two layers (z=-200m), mid velocity | 1000 | 1500 | - | - | |
| Model6 | Two layers (z=-200m), high velocity | 2000 | 3000 | | | |
| Model7 | Three layers ($z_1$=-200m, $z_2$=-400m) | 500 | 1000 | 1500 | - | |
| Model8 | Two layers (z=-200m), weak attenuation | Model 4 including $Q_S$=100, $Q_P$=200 | | | | |
| Model9 | Two layers (z=-200m), strong attenuation | Model 4 including $Q_S$=30, $Q_P$=60 | | | | |
| Model10 (Fig. S2) | Landau model (**real data validation**), four layers, no att. ($z_1$=-200m, $z_2$=-400m, $z_3$=-600m) | 450 | 750 | 900 | 1150 | |

## 2.2 Post-processing of the synthetic seismograms and comparison to analytical solutions

For each single simulation synthetic seismograms (or traces) are extracted at every receiver location in the synthetic borehole (gray lines in Fig. 2). The maximum amplitude for each trace (vertical component) is obtained to derive a frequency-dependent relation between signal amplitude and depth (red line in Fig. 2). The frequency-dependent amplitudes with depth are normalized to the amplitude at the surface. Finally, the interpolation of the resulting data shows the spectral amplitudes in dependency of the borehole depth (Fig. 3a).



As a benchmark, we compare the numerical results with two analytical solutions (Fig. 3b). The first solution (coloured interpolation in Fig. 3b) is based on a formulation of Barkan (1962)

$$A_z = \left(-0.2958e^{-\frac{(0.8474)2\pi z}{\lambda}} + 0.1707e^{-\frac{(0.3933)2\pi z}{\lambda}}\right) \tag{1}$$

where the amplitude of the vertical ground motion $A_z$ at depth z is a function of wavelength λ and z. The second analytical solution (dashed black lines in Fig. 3b) is the estimation of the Rayleigh wave penetration depth using various wavelength approximation (λ, λ/2, λ/3). For example, Hayashi (2008) and Kumagai et al. (2020) claim that surface wave penetration depth is down to a depth between *λ/4* and *λ/2*, whereas *λ/3* is often chosen to be the most suitable assumption (e.g., Larose, 2005). This approach is widely used to estimate the depth dependency of surface wave amplitudes in a homogeneous subsurface. The

analytical solutions are generally based on the interplay of seismic velocity $v$, frequency $f$ and wavelength λ:

$$\lambda = v/f \tag{2}$$

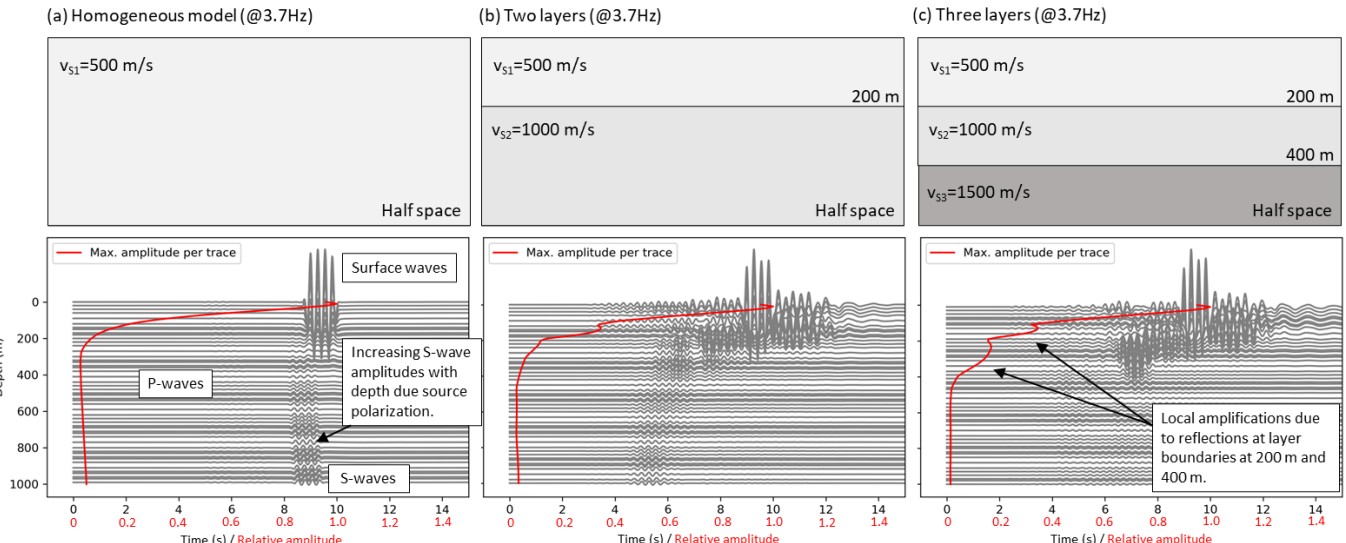

**Figure 2:** Example of the synthetic seismograms (gray lines) in dependency on the depth for signals at 3.7 Hz. The red line follows the maximum amplitude per trace and is affected by layers in the subsurface. It is normalized to the amplitude at the surface. P-, S-, and Rayleigh waves are simulated. The surface wave is dominating the wavefield near the surface.





## 3 Results

### 3.1 Homogeneous models

The comparison between analytical and numerical solutions (Fig. 3) applied to a homogeneous half-space model shows very similar results for the amplitude-depth relations per frequency. This implies that on the one hand the numerical simulation reliably reproduced the analytical calculations. One the other hand, an analytical solution might be sufficient, if the subsurface is approximately homogeneous. The estimation of the Rayleigh wave penetration depth fits very well to the more

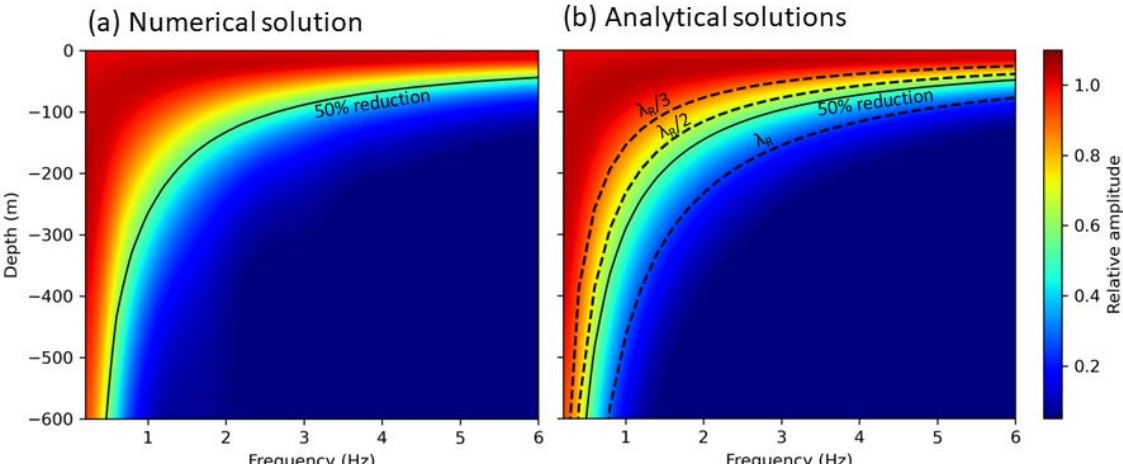

**Figure 3:** Benchmark - Comparison between numerical solutions and analytical solutions ($\lambda$ - estimations, dashed lines) for homogeneous models, based on the formulation in Barkan (1962). The results are very similar which proves the reliability of the numerical solution for this simple benchmark.

complex analytical solution (Fig. 3b); however, the fraction of $\lambda$ should be chosen, carefully considering the preferred reduction of noise with depth. These analytical solutions are limited regarding complex models of the subsurface.

Generally, a borehole should be deeper to yield a reduction of low frequency seismic noise (e.g., 1 Hz) compared to high frequencies (> 4 Hz). This is expectable, since the wavelength of a wave with a low frequency is larger compared to high frequencies. Consequently, the penetration depth of the surface wave is deeper. In view of eq. 2 the seismic velocity impacts this relation. The effects of the seismic velocity, signal frequency, layers in the subsurface, and attenuation on the depth-dependent amplitudes are simulated using homogeneous and layered models (Fig. 4). In case of high seismic velocities in the subsurface, deeper boreholes are required to yield a sufficient noise reduction. Furthermore, from the simulation results we obtain the effect of the signal frequency on the amplitudes. We find, for example, a borehole should be 100 m deep to reduce



the noise of 3 Hz signals in 4 km distance to the WT by 50 %, if the velocity of the S-wave is 500 m s$^{-1}$ (Fig. 4a), but the same borehole should be about 280 m deep for a velocity of 1500 m s$^{-1}$ (Fig. 4c).

### 3.2 Layered models

In case of a layered subsurface, we find that the amplitude decay with depth is dominated by the top layer which (here) has a thickness of 200 m. The comparison between Fig. 4d and Fig. 4g shows that a third deep layer with a high velocity has no significant impact on the results. However, again, the estimation of sufficient borehole depths depends strongly on the seismic

velocity of the layers (especially the top layer). A borehole with a depth of 200 m seems to be sufficient if the S-wave velocity of the top layer is approximately 500 m s$^{-1}$ (Fig. 4d), but this is not true if the velocity is increased (Fig. 4e,f). Signals > 4 Hz can by suppressed significantly in any case, but signals below this frequency (e.g., at 1 Hz) are not sufficiently affected by boreholes surrounded by rock with high seismic velocities. Hence, the geological setting and the seismic velocities play a key role concerning the evaluation of the effectivity of a borehole installation that aims to reduce the seismic noise produced by

WTs.

We further study the effect of attenuation (absorption) by specifying $Q_S$ and $Q_P$. In model 8 (Fig. 4h), we used relatively high Q values ($Q_S$=100 and $Q_P$=200) (Eulenfeld and Wegler, 2016) for a weak attenuation (e.g., compact rock) and in model 9 (Fig. 4i) we used relatively low Q values ($Q_S$=30 and $Q_P$=60) to simulate a strong attenuation (e.g., near-surface sedimentary rocks). We find that the general amplitude-depth relation is not significantly affected by attenuation compared with the same model

without attenuation (model 4). There are some frequency-dependent effects (e.g., at 4 Hz) showing slightly increased amplitudes below the depth of 200 m in case of strong attenuation (*A* in Fig. 4i). This can be explained by a reduced contrast between the amplitude at the surface and the amplitude in depth. A strong attenuation causes generally lower amplitudes compared to a scenario without attenuation; however, the contrast between the amplitude in the borehole relative to the surface seems to be weakened.





**Figure 4:** Effect of various models (a-c: homogeneous, d-g: layered, and h, i: with attenuation) on the frequency-dependent amplitude decrease with depth. The white dashed lines denote the layer boundaries. The solid black line indicates the amplitudes of 50 % reduction compared to the corresponding amplitude at the surface. The results are extracted from synthetic seismograms at 4 km distance from the source. Amplitudes as function of frequency are normalized to the corresponding amplitude at the surface.

## 3.3 Effect of distance between WT and seismic station




The frequency-dependent amplitude decay with depth is generally affected by the distance between the WT and the borehole.
To simulate these effects regarding the vertical and horizontal ground motion, we use model 4 (two layers with low velocities,
see Table 1) and decrease the distance between the source and the receivers systematically from 4 km to 1 km. The results are
presented in Fig. 5. With decreasing distance between the source and the borehole, amplitudes increase at frequencies between
2 Hz and 4 Hz up to a depth of 200 m, especially regarding the horizontal component in x-direction of the model. This indicates
relatively strong effects at the base of the topmost layer in 200 m depth, likely due to strong reflection concerning the specific
frequencies. These effects might change in case of higher velocities, change of frequency or thickness of the top layer.
Furthermore, we observe that the amplitude of the horizontal component is decreasing much faster with depth compared to the
vertical component. This behaviour can be described analytically (Barkan, 1962). However, layers in the subsurface can have
an adverse effect for specific frequencies, as described. The layer boundary in 200 m depth seems to isolate the amplitudes
above and below. This means that a borehole could be very effective at depths > 200 m, at least for this specific case.

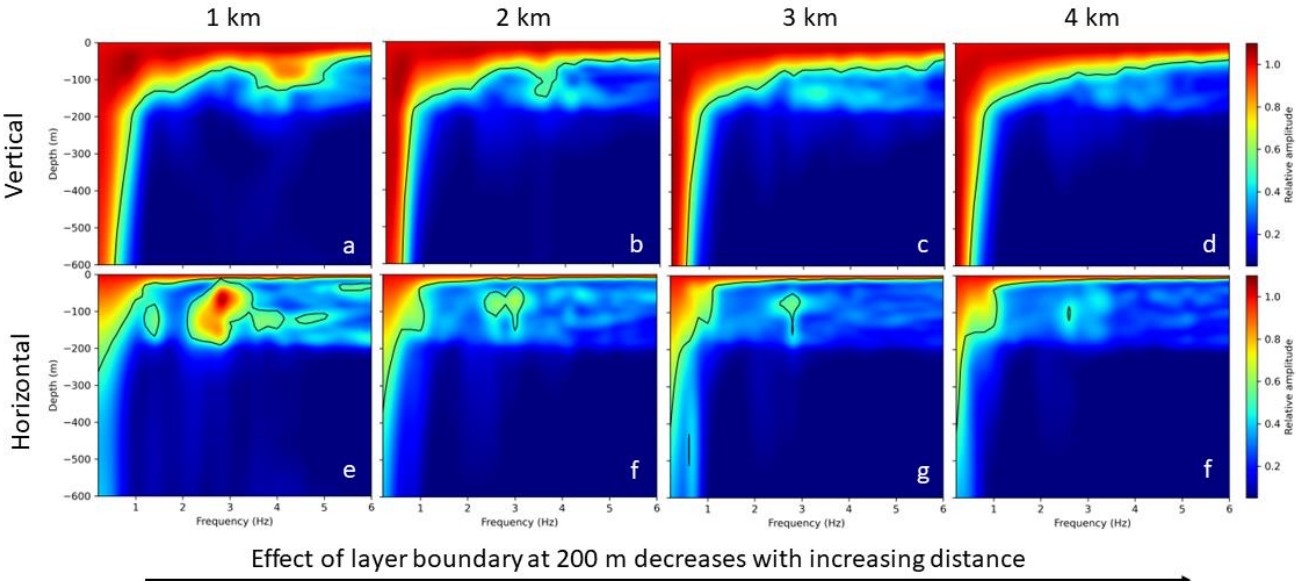

**Figure 5:** Effect of distance between source and borehole on vertical (z-axis) and horizontal (x-axis) seismogram components. Model 4 is used for these simulations. The distance has an effect on the amplitudes with depth, especially regarding the horizontal components. The layer boundary in 200 m depth is isolating the amplitudes above and below this layer. Amplitudes as functions of frequency are normalized to the corresponding amplitude at the surface.



## 3.4 Effects of attenuation

To investigate the effect of attenuation on the effectivity of a borehole for a specific frequency, we use model 4 including weak and strong attenuation. In this case, we study signals of 3.7 Hz (which is a typical frequency emitted by WTs) and calculated the seismic radiation in the x-z-plane and compare the results to those for the model without attenuation. As expected, we find that a strong attenuation affects the general amplitude decay with distance to the source and with depth (Fig. 6a, 6c, 6e). However, the relative amplitudes between depth and surface are only slightly affected by attenuation. This becomes obvious by looking at the almost identical results when the amplitudes in the depth are normalized to the corresponding amplitude at the surface (Fig. 6b, 6d, 6f). The tendency is that the contrast of amplitudes at the surface compared to amplitudes in depth is lower when strong attenuation exists (Fig. 6f). This implies that a borehole in a strongly attenuating environment might not be as effective as in less attenuating rock. However, the attenuation is not the dominating parameter to evaluate the effectivity of the borehole installation, as shown before. It should be noted that the undulation in x-direction is due to the layering (reflection effects).

With this analysis we can evaluate the distance of a seismometer to the WT. In view of Fig. 6c, we show that the distance between seismometer and WT could be reduced from 4 km to 2 km, if the seismometer is placed in a 100 m deep borehole, thus avoiding a significant increase of the noise level. But it should be clear that this is only an estimation for the specific case in this study and is very likely affected by changes of seismic velocity and the structure of the subsurface.

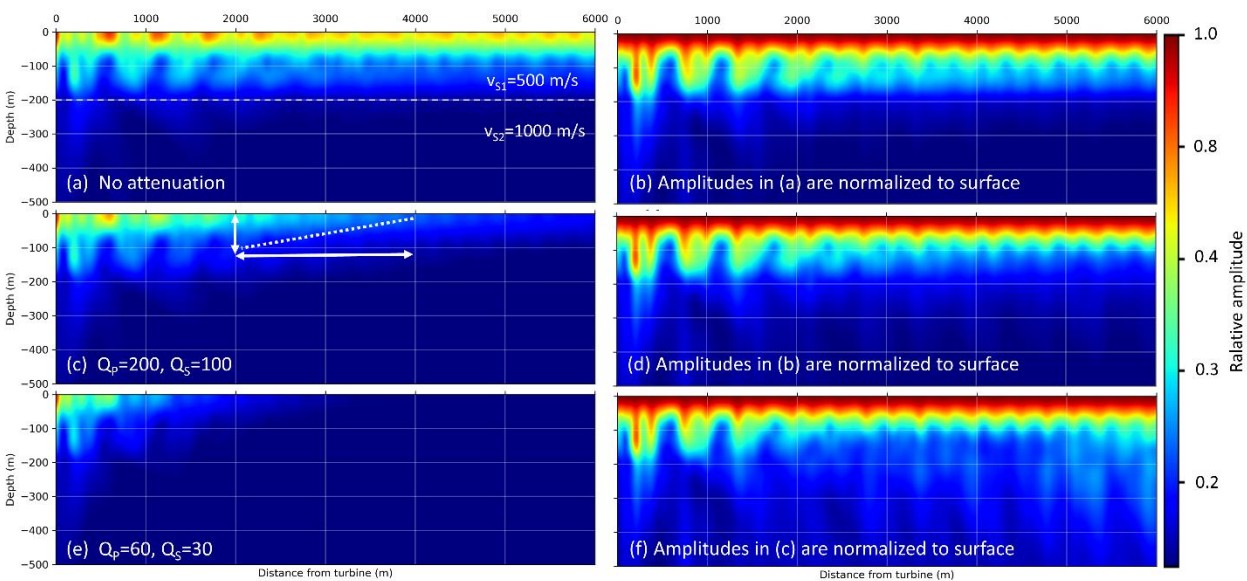

**Figure 6:** Left panel (a,c,e): Effect of attenuation on amplitude decays, normalized to source amplitude. Right panel (b,d,f): Amplitudes are normalized column wise, which means at each distance in x-direction, the amplitude with depth is normalized to the corresponding amplitude at the surface. The dominant signal frequency in these simulations is 3.7 Hz.



### 3.5 Real data validation

In this section, we validate the presented approach with data from seismic borehole installations. Close to the city of Landau in the upper Rhine valley, two seismic borehole stations with a depth of 305 m (station ROTT) and 150 m (station LDE) are located at distances of approximately 5.5 km (ROTT) and 3.8 km (LDE), respectively, to the next WTs (Fig. 7). These permanent stations are part of the earthquake monitoring system of the geological survey in Rhineland-Palatinate, Germany. Zieger and Ritter (2018) temporarily measured the frequency-dependent noise of the nearby WTs at the surface of the

corresponding borehole locations and calculated power spectral densities (PSD) (Fig. S3 in supplements). They showed a clear reduction of measured noise due to the boreholes. We took the PSD values of Zieger and Ritter (2018) and transformed the data into relative ground motions. At frequencies of 1 Hz we find an amplitude reduction of 73 % at the borehole station ROTT. At LDE we observe a reduction of 34 % for 1 Hz signals and 71 % for 3.7 Hz signals by comparing the amplitudes of the borehole seismometer with the surface amplitudes (Fig. S3 in supplements). The signals with 3.7 Hz are not reliably

observable at ROTT and are therefore not included in the further analysis. These factors of amplitude reduction are used as a reference for numerical results in our study. A numerical model (Fig. 8a) is built using subsurface information derived from the MAGS2 project (Fig. S2 in supplements) (Spies et al., 2017), which provides detailed seismic velocities and is hence one of the most accurate velocity models of the Landau region. The model of the local subsurface contains relatively low seismic velocities due to the younger sediments in the upper Rhine valley. The model we extracted has four layers with increasing S-

wave velocity from 450 m s$^{-1}$ (top layer), 750 m s$^{-1}$ (second layer), 900 m s$^{-1}$ (third layer) to 1150 m s$^{-1}$ (half space). Again, the density is fixed to 2600 kg m$^{-3}$ and the P-wave velocity is 1.7 times $V_S$. The synthetic boreholes in the numerical models are correspondingly located at distances of 3.8 km (LDE) and 5.5 km (ROTT), respectively, to the source point. From the



simulations (Fig. 8b and c), we can calculate the spectral isoline of an amplitude reduction of 73 % (ROTT) and the isoline of 71 % and 34 % (LDE) for all frequencies between 0.2 Hz and 6 Hz. Based on the model of the subsurface a comparison of the

numerical results with the observed data from Zieger and Ritter (2018) shows a good agreement and thus validates our amplitude estimations. The model is characterized by a first significant layer boundary at 200 m depth where the S-wave

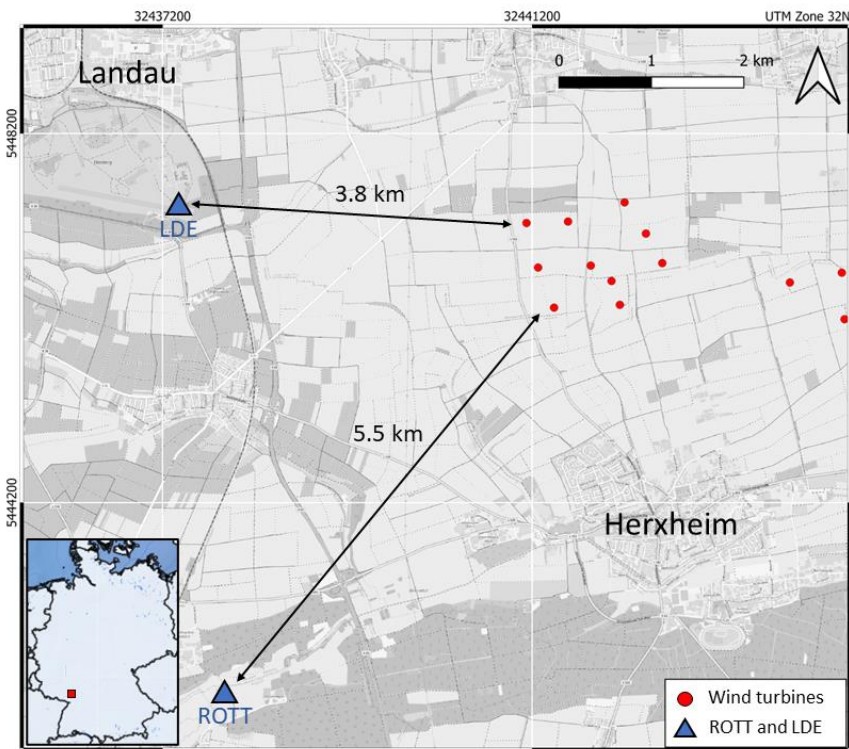

**Figure 7:** Map of the Landau region with borehole stations ROTT and LDE and the nearest wind farm north of Herxheim. Zieger & Ritter (2018) deployed two seismic stations at top of the boreholes to compare signal amplitudes measured at the surface with amplitudes measured at the borehole stations. (Maps: © OpenStreetMap contributors 2023. Distributed under the Open Data Commons Open Database License (ODbL) v1.0)

velocity increases from 450 m s$^{-1}$ to 750 m s$^{-1}$. Interestingly, this layer boundary significantly affects the amplitude decrease with depth, especially regarding signals with a frequency between 2 Hz and 4 Hz. This effect is likely due to reflections of the waves that are mainly traveling along the top layer. Considering these effects, the observed amplitude reduction by 71 % of

the 3.7 Hz signals can only be reproduced numerically by the layered model and would fail for a homogenous model. The observed reduction by 34 % of the 1 Hz signals at borehole station LDE is also accurately described by our modelling. The reduction by 73 % of the 1 Hz signals at ROTT is simulated appropriately. However, there is a discrepancy between observed and simulated amplitude reductions at a depth of 305 m (ROTT). In this frequency range (around 1 Hz), the amplitude decay with depth is very sensitive and thus challenging to be perfectly fitted.





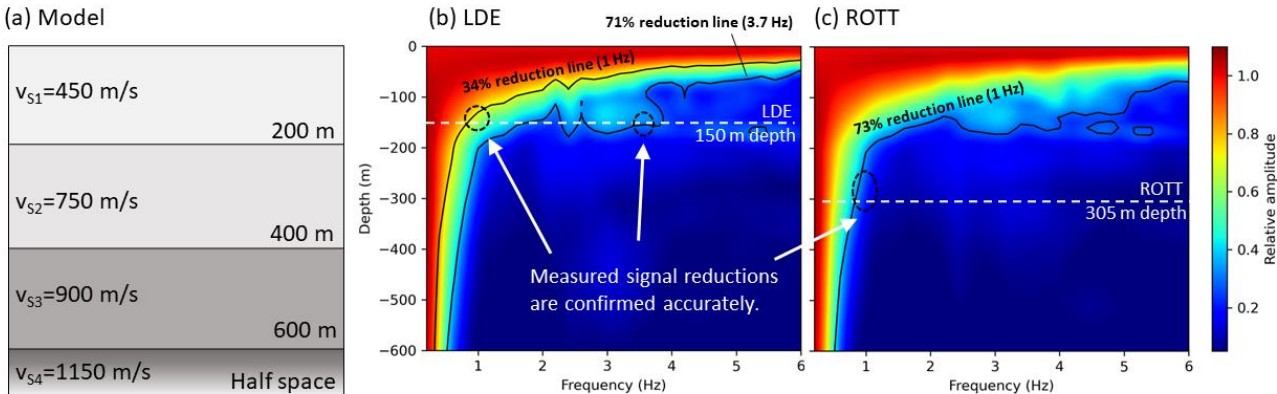

**Figure 8**: a) A model with three layers above a half space is used for the real data validation. The model is based on information provided by the MAGS2 project (Spies et al., 2017). The results of the simulations are compared to observations made by Zieger and Ritter (2018) and show good agreement, this means that the reduction of noise amplitudes can be reliably estimated using 2D numerical simulations.

## 4 Discussion and conclusions

In this work, we study the effectivity of borehole installations to reduce the impact of seismic noise produced by WTs on
seismological recordings. Based on numerical models, the effect of geophysical parameters, such as seismic velocities and attenuation, and layering of the subsurface are simulated to constrain the depth of seismic borehole stations, to significantly reduce the noise produced by WTs?

We validate our approach by comparisons with existing real data published by Zieger and Ritter (2018). We can reproduce the observed reduction factors by Zieger and Ritter (2018) of signal amplitudes at specific frequencies measured at the surface
and in depth at two boreholes close to Landau in Rhineland-Palatinate, Germany (Fig. 8). We point out that this validation is based on simulations using a realistic model of the subsurface which consists of three layers above a half space (based on results given in Spies et al., 2017). Interestingly, we would not be able to explain some of the observations, if the layer boundary at 200 m would not be included in the model. This indicates that simplified analytical solutions (homogeneous half-space model) fail to simulate the wavefield sufficiently. To increase the reliability and to enable a wider application of the method,
further borehole data, covering a broad range of frequencies, is necessary and should be studied in the future. Our real data validation is performed for the upper Rhine valley which is characterized by thick relatively young sediments with low seismic



velocities (Fig. 8a). Similar simulations could be performed for other geological settings characterized by more compact rock types.

The numerical modelling shows that the effectivity of such boreholes to reduce surface generated seismic noise strongly depends on the interplay of signal frequency, seismic velocity and the wavelength (Fig. 4). Low-frequency signals and high seismic velocities yield a large wavelength, which results in a penetration depth of > 600 m for the most prominent surface waves. In regions with soft sediments, boreholes of a few hundred meters depth are likely effective to reduce the noise from WTs, especially in view of the high-frequency signals. A borehole of only 200 m to 300 m depth can reduce the noise of signals between 2 Hz and 6 Hz by more than 50 %. However, boreholes might not be effective in other regions where more compact rock types and relatively high seismic velocities dominate. The typical frequency range of signals produced by WTs is between 1 Hz and 10 Hz. The reduction of signals with frequencies around 1 Hz seems challenging, due to the relatively large wavelength. These waves travel generally very far in distance and depth. Nevertheless, Zieger and Ritter (2018) demonstrated the reduction of such signals by 73% in a borehole of 305 m depth (5 km to the next WF) and 34 % in case of 150 m depth (3.8 km to the next WF). We confirmed these observations with our modelling (Fig. 8).

The comparison of results for homogeneous models and layered models shows that the amplitude-depth relation is dominated by the top layer, but this depends again on the general wavelength of the surface waves and the thickness of the top layer (Fig. 4). We studied these effects for a top layer of 200 m thickness, which is characteristic for the Upper Rhine valley. The effects of various thicknesses and lateral heterogeneities (such as fault structures or site effects) could be part of future modelling studies.

We further show that borehole installations in geological environments with strong attenuation might not be effective as in environments with weak attenuation. Attenuation reduces the amplitude with distance in general, but it does not affect the relative amplitudes at the surface and in depth significantly (Fig. 6).

We show that the effects of the layer boundary at 200 m depth on the wavefield increases with decreasing distance to the source, especially regarding the horizontal components of the signal (Fig. 5). In our simulations we apply vertically polarized source mechanisms to model the signals from the WTs. This is an approximation to the up and down movement of the foundation of the WT. However, other source mechanisms and polarizations might have additional effects on the wave propagation and should be part of future research. A time-limited wave package is a sufficient approximation of the source signal and a practical solution to numerically simulate effects of the subsurface on the wave propagation. However, WTs usually emit continuous signals which might lead to additional complex wave reflections and interferences in the subsurface. Further signal modulation can also occur by wavefield interferences from multiple WTs, as shown by Limberger et al. (2022). A key aspect in evaluating the effectivity of a borehole is the general purpose of a specific seismic station. If a station is used for the detection and localization of local earthquakes with a relatively high frequency content (e.g., higher than 5 Hz), a borehole can be very effective to reduce the noise from WTs nearby. However, if the seismometer is used to measure signals with lower frequencies (e.g., 1 Hz in case of teleseismic signals), then borehole installations might fail in reducing the noise, or the necessary borehole would require a depth that is too large to be feasible. In view of our results, we strongly recommend



to perform estimations based on the specific characteristics of the location of interest and not to generalize and apply one estimation for all locations and seismic stations. However, besides WTs, our approach can be also applied to other anthropogenic noise sources (e.g., in urban areas) and enables a universal assessment of seismic noise and its effect on borehole installations.

To conclude, the impact of seismic noise produced by WTs on seismometers can be decreased if the seismic sensor is installed within a borehole at an adequate depth. But this strongly depends on various geophysical and geological parameters, such as seismic velocities or layering in the subsurface, and should be carefully evaluated for every geological environment separately. With this study, we provide a robust approach to perform reliable estimations of the effectivity of borehole installations.

**Code and data availability.** The numerical simulations were performed using the commercial software package Salvus (Afanasiev, 2018). The simulation scripts are available from the corresponding author (limberger@igem-energie.de) on request. The data processing was performed using the Python packages NumPy and SciPy.

**Author Contributions.** All authors developed the thematic idea of the study. F.L. performed the numerical simulations and processed the data. G.R. participated in developing the model, and supervised the writing of the article. F.L., M.L., and G.R. interpreted the results. F.L., G.R., M.L. and H.D. each edited the article.

**Competing Interests Statement.** The authors declare no competing interests.

**Acknowledgements.** We thank the German Federal Ministry for Economic Affairs and Climate Action (FKZ no. 0324360) and ESWE Innovations und Klimaschutzfonds for their support in performing this study.

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
