# Peer review of "The impact of seismic noise produced by wind turbines on seismic borehole measurements"

_EGUsphere, 2023_

## Referee Comment (RC2)

Limberger et al. use numerical models to simulate the effect of wind-turbine generated noise on seismic stations installed in boreholes. In particular, they model the Landau region geology and nearby wind farms to validate their results against existing measurements from Zieger and Ritter (2018). The study is well structured and the effect of wind turbine noise is systematically analysed for different sensor depths, geological layering and associated seismic velocities, and damping parameters of the subsurface, requiring minor revision. However, I am missing some explanation on the authors' choices of the input and presented results in several parts of the manuscript. Also, by giving recommendations on how to apply the model for designing new borehole stations in existing settings, where seismic monitoring is necessary and wind farms are present. These suggested changes would make the study more practical and give the work more impact. For example, this could be addressed i) by stating the performance of the modelling in the more complex Landau setting and ii) by showing which settings (number of layers, dimensions, etc) can be simulated and which not. Also, it would be nice to show for which borehole depth the gain of placing borehole sensors is highest.

l. 57: The authors should describe in more detail why this source signal was chosen. In particular, is this an average signal of WT noise due to strong winds or a specific tower height? What controls the frequency content? This could be explained with respect to the presented results of Zieger and Ritter (2018), that classify strong versus weak noise conditions.

l. 158ff: This section is a theoretical approach but not very useful. It would be much more applicable if a more realistic attenuation could be modelled, as shown e.g. by Bethmann et al. (2012) https://doi.org/10.1111/j.1365-246X.2012.05555.x or a gradient rather than just 2 layers with constant values. Attenuation is important to consider in the analysis but should not be too simplified. This is similar to my suggestion of stating the complexity of settings that could still be simulated with the model and should be included in the discussion.

l. 258: Maybe a similar figure as Fig 2 of Prevedel et al. (2015) https://doi.org/10.1007/s00531-015-1147-5 could be attempted basyed on the modelling results of wind turbine noise.

4 Discussion and conclusions:
How difficult would it be to include other measures in the modelling (e.g. trenches as given by FA Wind (2021), Minderung seismischer Wellen von Windenergieanlagen. Strukturelle Maßnahmen auf dem Wellenweg.)?

As the authors state themselves the study would benefit from more data for validation.

Minor comments:
l. 54 of the complete wavefield

Fig. 2: The red line in Fig. 2 requires a more detailed description and explanation in the text. Should the red labelling be displayed on the vertical axis? What controls the amplitudes of the synthetic traces? Are the relative amplitudes normalised to the input signal? Please explain better the meaning of this figure.

Fig 5: Show the reference values from Model 4 (e.g. black line in Fig 4 d) by dashed lines in this figure.

---

## Author Comment (AC1)

**Comment on egusphere-2023-45 by Sven Schippkus Referee #1**

We sincerely thank Sven Schippkus for the valuable comments which will help to improve the paper. Please find a point-by-point response to the comments. Our responses are in red. Actual changes to the manuscript are written in *italic.*

On behalf of all authors, Yours sincerely,

Fabian Limberger

**Comments by Sven Schippkus Referee #1:**

In the manuscript entitled "The impact of seismic noise produced by wind turbines on seismic borehole measurements" by Limberger et al., the authors present a parameter study on the impact local geology and experiment geometry have on recordings of seismic surface waves, which were emitted by wind turbines. These tests are performed using Salvus, a well-established spectral element wave propagation simulator. The goal of these tests is to give insights into and guidelines for required borehole depths for sufficient suppression of surface waves. The manuscript is a valuable contribution to the ongoing discussion on approaches to accommodate the increasing societal need for wind turbines in the operation of seismological observatories.

The manuscript is in a mature state and I don't see any major issues. While not all source effects are accounted for in the authors' description of the wind turbine source, I believe the fairly straightforward description they choose is likely sufficient to support the conclusions the authors draw from the simulations. I don't think further simulations are necessary, but a few minor changes to figures, some wording, and the discussion would improve the accessibility of the manuscript. I believe addressing my comments below constitutes a minor revision.

A) Wind turbine noise. The fundamental benefit of borehole installations is implicit throughout the text but never explicitly stated. The desire for a reduction of surface wave noise through amplitude reduction of all surface waves implies that body waves carry the signal of interest. Surface wave "signals" are impacted to the same extent as "noise" in terms of relative amplitude reduction. To me this appears to be one of the major reasons other approaches to wind turbine noise reduction, which the authors introduced briefly in the introduction, have significant merit. At least some of them aim to differentiate surface waves from particular sources from other surface waves, e.g, in ML-based denoising. I'd appreciate a discussion of this aspect by the authors to gain insight into their thoughts on the position of the approach in this study relative to the overarching problem of wind turbine noise in seismic recordings and which direction they deem as most promising and why.

> We agree with the referee's comment; if the seismic station is purposing to detect surface waves, then a borehole station might have an adverse effect, because the signal is also reduced in amplitude. However, in terms of earthquake detection, especially regarding local (e.g., geothermal sites) and regional seismic events a borehole can improve signal/noise ratio significantly. Compared to other measures (e.g., denoising based on machine learning approaches) the waveforms are not affected, which might be a problem concerning conventional and modern filter techniques. Nevertheless, a combination of both, seismometer in boreholes and advanced filter techniques could be sufficient, depending on the situation.

We did not include this aspect in the discussion and will add the following sentences to the discussion section (in Line 263).

*Obviously, if signals from surface waves are to be measured, a borehole would not be the appropriate choice to reduce the impact from wind turbines, however, in this case alternative techniques such as noise filters based on machine learning (e.g., Heuel & Friederich, 2022) could help to increase the signal quality. Generally, filter techniques could affect the waveform and signal amplitude of the desired signal and should be considered carefully concerning their application. Combinations of both sensors in boreholes and advanced filter techniques could also be considered.*

B) The wind turbine source. The authors choose a simple description of the wind turbine source, a vertically acting sinusoidal force. While the authors discuss some of the drawbacks and potential for future improvement of this choice, a few questions remain. A vertically acting force produces no Love waves, and they then may only emerge due to wave type conversion. To me, it appears more likely that the majority of movement of the wind turbine tower base is horizontal, in accordance with the eigenmodes of the tower (which correspond well to the spectral peaks a wind turbine produces). The authors implicitly acknowledge this, but I wonder what the impact on the results would be. Would this really matter for the intended message of this manuscript? Maybe not, I'm also not sure.

That is an interesting point. The movement of the foundation of a wind turbine is very complex and may involve tilting, up-down, circulating, and horizontal motions. Probably there could be an effect on the emitted waves, which should be studied in future, but it is not within the scope of this work. In our study, we focus on amplitudes variations between the surface and depth. Whether this amplitude is caused by a tilting motion, or up and down motion of the wind turbines is of secondary interest in our study, which is why we decided to choose the approximation of an up- and down motion for simplification.

Additionally, in the considerations here, all frequencies are treated as equally important (because only relative amplitude reduction at each frequency is investigated). Wind turbines are known to dominantly generate certain frequencies, as the authors also utilise for their relative amplitude reduction estimation on field data. This brings up two questions for me: What is the benefit of repeating the simulations for each frequency (with a tapered sinusoid) instead of formulating a source term more comprehensively, e.g., a sum of harmonics with high amplitudes only for those harmonics corresponding to wind turbine eigenmodes (and blade passing frequency)? Because certain frequencies dominate, the issue of wind turbine noise overpowering body wave signals is worse for those frequencies. What is the authors' view on this?

We agree that a more advanced source term (e.g., superposition of signals) could cover the typical frequencies emitted by a wind turbine. There are many types of wind turbines emitting signals at various frequencies between approx. 1 and 10 Hz. Therefore, it is not certain that a wind turbine always emits at 1 Hz, for example. These frequencies might be slightly shifted depending on the type and dimension (e.g., tower height) of the turbine. We wanted to give a broader view on the spectra, especially

concerning lower frequencies, since these waves reach very far and are critical. Furthermore, to avoid a necessary filtering in the postprocessing of the synthetic seismograms and to avoid complex effects of signal interferences of within frequencies, we considered separate simulations per frequency.

C) Figure 3: Two aspects: 1) Maybe it could be helpful to quantify the mismatch between a) and b), at least for the first analytical prediction (colored background). Currently, the authors describe the agreement between them as "very good". I tend to agree but it would give more confidence to quantify this. This could be done by computation some similarity measure, or maybe by showing a difference plot between the two. I'd expect them to be not exactly the same and it would be interesting to see (and discuss?) what differences emerge and why.

We thank the referee for this recommendation. We added the difference plot and discuss the outcome in the supplements. The difference is not zero at relatively surficial areas for high frequencies and in deeper regions for low frequencies. This indicates that the discrepancy between analytical and numerical solution is wavelength dependent. However, considering that the maximum difference is about 0.1 in amplitude, the numerical and analytical solution are in good agreement. The numerical approach includes body waves, which are completely neglected in the analytical approach. This could be one explanation for the slight differences.

[Figure]

2) The dashed lines corresponding to lambda, lambda/2, lambda/3 were confusing to me at first. I think it would be helpful to expand in the text a bit more on the meaning of these lines. In some way, each of these lines represents an entirely different distribution (colored background) where amplitude is 1 above the line, and 0 below. At least, that's how I understand what is colloquially meant by the term "penetration depth". Of course, we know that there is no cut-off depth for surface waves in that sense, but it could be helpful for the reader to state that in your model a lambda/3 penetration depth means relative amplitudes are still 90% (or whatever the corresponding value exactly is, and the other wavelengths with lower values) to make the connection clearer.

To clarify, we extended the text (line 96) by:

*Usually, a fraction of lambda (lambda/2 or lambda/3) is used to estimate the penetration depth of surface waves. However, this is likely underestimating and neglecting the amplitudes at depth. E.g., lambda/3 exhibits a reduction of about 10% and lambda/2 of 30%, which implies that this simple approximation is inadequate to derive the frequency-dependent amplitudes as function of depth.*

D) Figure 8: The representation of the data points extracted from field data could be a bit more precise. Currently, the measured amplitude reductions (34%, 71%, 73%) are marked by what appear to be hand-drawn dashed markings. The exact mismatch (which is likely very low) does not become clear. Maybe it would be helpful to plot a colored circle (with the color corresponding to the field data measurement) on top of the simulation results in the background to give a better visual indication of how well-matched they are. I found the lines confusing at first, because they are measured for one frequency (on data) and then expanded to all frequencies (from modelling). The exact relation between the two does not become clear.

We agree. The isolines will be removed to avoid misunderstandings and make the plot cleaner and better to understand. Instead, we added crosses and pluses to mark the corresponding points.

Before:

[Figure]

After:

[Figure]

E) Colors. I recommend to use accessible colormaps instead of jet for all figures. For more details on why see Crameri, F., Shephard, G.E. & Heron, P.J., 2020. The misuse of colour in science communication. *Nature Communications*, 1–10. doi:10.1038/s41467-020-19160-7.

> We thank the reviewer for the recommendation and see the point. Compared to e.g., batlow, jet provides a strong contrast within the colormap that is very useful to show small difference of amplitudes, which is of advantage in our case (see figure below). Therefore, we would like keep jet in this case; however, we will use more accessible colormaps in the next manuscript.

[Figure]

F) Author credits. While I don't know whether EGUsphere gives specific rules for author attributions, the authors may find the CRediT system useful to cover all relevant aspects (https://credit.niso.org).

> Thank you for this advice.

---

## Author Comment (AC2)

**Comment on egusphere-2023-45 by Anonymous Referee #2**

We sincerely thank the referee for the valuable comments which will help to improve the paper. Please find a point-by-point response to the comments. Our responses are in red. Actual changes to the manuscript are written in *italic.*

On behalf of all authors, Yours sincerely,

Fabian Limberger

**Comments by Anonymous Referee #2:**

Limberger et al. use numerical models to simulate the effect of wind-turbine generated noise on seismic stations installed in boreholes. In particular, they model the Landau region geology and nearby wind farms to validate their results against existing measurements from Zieger and Ritter (2018). The study is well structured and the effect of wind turbine noise is systematically analysed for different sensor depths, geological layering and associated seismic velocities, and damping parameters of the subsurface, requiring minor revision. However, I am missing some explanation on the authors' choices of the input and presented results in several parts of the manuscript. Also, by giving recommendations on how to apply the model for designing new borehole stations in existing settings, where seismic monitoring is necessary and wind farms are present. These suggested changes would make the study more practical and give the work more impact. For example, this could be addressed i) by stating the performance of the modelling in the more complex Landau setting and ii) by showing which settings (number of layers, dimensions, etc) can be simulated and which not. Also, it would be nice to show for which borehole depth the gain of placing borehole sensors is highest.

l. 57: The authors should describe in more detail why this source signal was chosen. In particular, is this an average signal of WT noise due to strong winds or a specific tower height? What controls the frequency content? This could be explained with respect to the presented results of Zieger and Ritter (2018), that classify strong versus weak noise conditions.

The chosen frequency range of the source signal (0.2 Hz-6 Hz) is typical for wind-turbine induced signals, as explained in line 66. Especially, waves with a relatively low frequency (about 1 Hz) are widely observed at seismometers in the neighborhood of wind turbines. The frequencies of about 1 Hz and 3.7 Hz are typically related to bending modes of the tower. We agree that, depending on the wind turbine type, these frequencies might be slightly shifted from turbine to turbine. To keep the estimations and methods universal, we have chosen to use all frequencies between 0.2 Hz and 6 Hz as source signals, instead of specific ones.

We add the following text to line 67 for clarification:

> *Signals at about 1 Hz and between 3 Hz and 4 Hz are widely observed by seismometers close to WTs. These frequencies are related to the tower eigenmodes of a WT (Zieger & Ritter, 2018; Zieger et al., 2020) and depend on the type and specifications of the WT. Hence, instead of choosing just a few specific frequencies corresponding to one specific wind turbine type, we keep the approach universal and study various frequencies between 0.2 Hz and 6 Hz.*

l. 158ff: This section is a theoretical approach but not very useful. It would be much more applicable if a more realistic attenuation could be modelled, as shown e.g. by Bethmann et al.

(2012) https://doi.org/10.1111/j.1365-246X.2012.05555.x or a gradient rather than just 2 layers with constant values. Attenuation is important to consider in the analysis but should not be too simplified. This is similar to my suggestion of stating the complexity of settings that could still be simulated with the model and should be included in the discussion.

> That is an interesting point. We agree that in some cases, the chosen attenuation model should be more complex including more layers with attenuation or even a gradient. However, with our simulations we find that the attenuation is not the key parameter that controls the amplitudes as function of depth. Therefore, we did not go into further detail concerning attenuation. We wanted to look systematically at various effects, starting with simple models. In general, using the Mondiac-software package Salvus (as we did), models can be designed with high complexity including a large number of layers, parameter gradients etc. This means that the limit of complexity is rather defined by the available amount of subsurface data, than by the software that is used.

l. 258: Maybe a similar figure as Fig 2 of Prevedel et al. (2015) https://doi.org/10.1007/s00531-015- 1147-5 could be attempted based on the modelling results of wind turbine noise.

> We see the advantage of such a figure. However, the analysis of signal to noise ratio would require the simulation of a target signal (such as earthquake signals) as well, which is beyond the scope of our study, but could be an important step in the future.

4 Discussion and conclusions: How difficult would it be to include other measures in the m odelling (e.g. trenches as given by FAWind (2021), Minderung seismischer Wellen von Windenergieanlagen. Strukturelle Maßnahmen auf dem Wellenweg.)? As the authors state themselves the study would benefit from more data for validation.

> That's a very good point. As mentioned above, using the software package Salvus, models can be designed to be very complex. This means that trenches, filled caverns, or other structural aspects (e.g., topography) could be included in the simulation. We agree that this is an important point to be considered in the future to effectively reduce noise from wind turbines. However, within our study we wanted to focus on general ("natural") effects of the subsurface, such as layering and velocity parameters. The incorporation of structural measures (see Abreu et al., 2022) and borehole installations could be a promising solution and should be studied in further research.

> We added a sentence to the discussion (line 246) to specifically point to that aspect:

>> *Moreover, additional structural measures (e.g., filled trenches) as studied by Abreu et al. (2022) could be included in the simulation to incorporate the noise-reducing effects due to boreholes as well as structural measures.*                    .

**Minor comments:**

l. 54 of the complete wavefield

This mistake is corrected.

Fig. 2: The red line in Fig. 2 requires a more detailed description and explanation in the text. Should the red labelling be displayed on the vertical axis? What controls the amplitudes of the synthetic traces? Are the relative amplitudes normalised to the input signal? Please explain better the meaning of this figure.

We extended the caption of the figure to clarify the normalization and "how-to-read" the figure.

Fig 5: Show the reference values from Model 4 (e.g. black line in Fig 4 d) by dashed lines in this figure.

Thank you for this suggestion. We added the 50% reduction line from Fig. 4d (which is actually identical to Fig. 5d) as a dashed line in all other subfigures in Fig. 5.